# Preventing and Treating Pain and Anxiety during Needle-Based Procedures in Children with Cancer in Low- and Middle-Income Countries

**DOI:** 10.3390/cancers16051025

**Published:** 2024-03-01

**Authors:** Michael J. McNeil, Ximena Garcia Quintero, Miriam Gonzalez, Yawen Zheng, Cecilia Ugaz Olivares, Roxana Morales, Erica Boldrini, Débora Rebollo de Campos, Daiane Ferreira, Kamalina Coopasamy, Joliza Caneba, Maria Louisa Padernilla, Stefan Friedrichsdorf, Justin N. Baker, Paola Friedrich

**Affiliations:** 1Department of Global Pediatric Medicine, St. Jude Children’s Research Hospital, Memphis, TN 38105, USAmiriam.gonzalez@stjude.org (M.G.);yawen.zheng@stjude.org (Y.Z.);; 2Division of Quality of Life and Palliative Care, Department of Oncology, St. Jude Children’s Research Hospital, Memphis, TN 38105, USA; 3Instituto Nacional de Enfermedades Neoplásicas, Lima 15038, Peru; ceciugaz1@gmail.com (C.U.O.); roxana_dm16@hotmail.com (R.M.); 4Hospital de Câncer Infantojuvenil de Barretos, Barretos 14784-400, Brazilu01593@hcbinfantil.com.br (D.R.d.C.); u06010@hcbinfantil.com.br (D.F.); 5Grey’s Hospital, Pietermaritzburg 3201, South Africa; 6Philippines General Hospital, Manila 1000, Philippines; 7Division of Pediatric Pain, Palliative Care & Integrative Medicine, University of California San Francisco, San Francisco, CA 94143, USA; 8Division of Quality of Life and Pediatric Palliative Care, Stanford University, Palo Alto, CA 94305, USA

**Keywords:** quality of life, pediatric oncology, procedural pain

## Abstract

**Simple Summary:**

Children with cancer experience pain and anxiety during needle-based procedures. Undertreated pain in children has long-lasting consequences and reduces the efficacy of subsequent analgesic efforts. A validated quality improvement (QI) intervention has been shown to be effective in reducing pain and anxiety during needle-based procedures in children but has only been implemented in high-income contexts. However, most children with cancer live in low- and middle-income countries (LMICs), and many do not have access to central venous access and require dozens to hundreds of needle-based procedures throughout their treatment. This validated QI intervention, known as the “Childrens’ Comfort Promise”, was successfully adopted, adapted, and implemented in four global hospitals caring for children with cancer around the world. This multi-site, multi-country QI initiative feasibly resulted in improved procedural pain in children, with high satisfaction of both healthcare professionals and patients/families. Further work is needed to overcome the challenges of educating the workforce and achieving consistent access to supplies such as topical anesthetics. Additional plans include modifying the Global Comfort Promise to include high-quality communication and expanding to additional sites with further refinement of the implementation strategy.

**Abstract:**

*Introduction*: Children with cancer experience significant pain and anxiety during needle-based procedures. Undertreated pain in children has long-lasting consequences and reduces the efficacy of subsequent analgesic efforts. A validated quality improvement (QI) intervention, known as the “Children’s Comfort Promise”, includes (1) topical anesthetics, (2) sucrose or breastfeeding for infants, (3) comfort positioning, and (4) distraction techniques, and has been shown to be highly effective in decreasing procedural pain and anxiety in children. However, there is limited data about the adoption, adaptation, and implementation of these interventions in low- and middle-income countries (LMICs). *Methods*: A QI pilot project utilizing the Model for Improvement of the “Global Comfort Promise” was implemented in four global pediatric cancer hospitals (Lima, Peru; Barretos, Brazil; Pietermaritzburg, South Africa; and Manila, Philippines). Between August 2021 and January 2023, the pilot sites identified a specific aim, co-designed the measurement strategy with St. Jude Children’s Research Hospital, and adopted, adapted, and implemented the project at their individual sites. *Results*: A total of 2,185 different procedures were recorded in the first year of implementation. Most patients were less than 10 years old (60.5%) and solid tumors (37.9%) were the most common diagnosis. Overall, healthcare professionals (98.3%) were satisfied with the procedures. Parents and patients reported that only 33.7% of patients experienced pain during the procedure. All (100%) parents and patients felt the healthcare teams adequately addressed their child’s pain. Median self-reported adherence to ≥2 interventions was 98.0%. Challenges to the implementation of the QI initiative included lack of training, turnover of the medical staff, maintaining staff enthusiasm, and access to topical anesthetics. Each site had unique change ideas to implement the initiative. *Conclusions*: This multi-site, multi-country QI initiative was feasible and was successfully adopted, adapted, and implemented in the LMIC context to improve procedural pain in children (Global Comfort Promise). Additionally, this intervention resulted in high satisfaction of both healthcare professionals and patients/families. Further work is needed to overcome the challenges of topical anesthetic access and education of the workforce. Additional plans include modifying the Global Comfort Promise to include high-quality communication and expanding to additional sites with further refinement of the implementation strategy.

## 1. Introduction

In 2010, the Declaration of Montreal affirmed that access to adequate pain management is a fundamental human right [1]. Despite this declaration, research demonstrates that pain in pediatric patients is common, underrecognized, and undertreated [2,3,4]. Additionally, for parents whose children are hospitalized, their second highest priority, after getting the diagnosis right, is to adequately care for their child’s pain [5], and their greatest distress is failing to control their child’s pain [6,7]. Undertreated pain in children has significant consequences throughout the child’s life, including increased morbidity and mortality [8,9]. Additionally, repeated painful procedures in infants and children induce changes in long-term pain sensitivity and reduce the efficacy of subsequent analgesic efforts [10,11].

A survey at a large US-based children’s hospital showed that the most common source of pain and anxiety in children were needle-based procedures such as lab draws or peripheral intravenous catheter (PIV) insertions [3]. However, in a comparative chart review, there was a pronounced discrepancy between what was reported by the children and parents and what was recorded in the medical chart [3]. While the majority of research on pediatric pain occurs in North America and Europe, a comparable study performed in South Africa found similar results, with needle-based procedures being the most common source of pain, while only 16% had documented pain scores [12].

To address this deficit, Friedrichsdorf et al. published the first report of a successful “Children’s Comfort Promise” intervention to reduce or eliminate pain during needle-based procedures. The protocol implemented consistent use of (1) topical anesthesia (4% lidocaine cream), (2) sucrose/breastfeeding for infants aged 0–12 months, (3) positioning (including swaddling, skin-to-skin, or facilitated tucking for infants; sitting upright for children >6 months), and (4) age-appropriate distraction [13]. The multi-site quality improvement (QI) intervention was established system-wide in one of the largest children’s hospital systems in the United States using a staggered implementation approach over a 3-year period to allow for unit-specific customization and facilitation of knowledge transfer from one unit to another. All departments were required to offer all four strategies with appropriate education at least 95% of the time. A comparison of baseline audits with continuous post-implementation audits revealed that wait times for services decreased, patient satisfaction increased, and staff concerns about implementation were allayed (e.g., concerns about wait times and success rates of venipuncture after topical anesthesia) for more than 200,000 needle procedures/year [13]. Subsequently, the Comfort Promise has been implemented in children’s hospitals in the United States, Canada, Denmark, and Uruguay as a standard of care of preventing and treating pain and anxiety caused by needle procedures.

However, little work has been done to address pain from needle-based procedures for children in LMICs, especially children with cancer. While dramatic improvements have been made in improving outcomes for pediatric malignancies in high-income countries (HICs), almost 90% of children with malignancies live in LMICs, and many will die of their disease [14,15,16,17]. Compounding this is the fact that many health systems in LMICs do not have capacity for central venous lines (CVLs), resulting in children with cancer requiring dozens to even hundreds of needle pokes during their treatment course. The avoidance of repeated painful procedures has been identified as a cause of early treatment abandonment and mortality in LMICs [18].

To assess the feasibility of implementing this QI in LMICs, the “Global Comfort Promise” was implemented at four culturally unique institutions around the world with varied resource capabilities (Hospital de Amor in Barretos, Brazil, Instituto Nacional de Enfermedades Neoplásicas in Lima, Peru, Philippines General Hospital in Manila, Philippines, and Grey’s Hospital in Pietermaritzburg, South Africa) [19,20]. The purpose of this report is to describe the adoption, adaptation, and implementation of the “Global Comfort Promise” at these institutions utilizing the Model for Improvement to improve the experience of needle-based procedures for children with cancer in resource-constrained settings around the world [19]. This manuscript follows the SQUIRE 2.0 guidelines for reporting a QI initiative [20].

## 2. Methods

### 2.1. Context

St. Jude Global (a key strategic initiative of St. Jude Children’s Research Hospital) is a global program that brings together individuals and institutions dedicated to the shared vision of improving the quality of healthcare delivery and increasing survival rates for children with cancer and blood disorders worldwide through the development of capacity-building, educational, and research initiatives [21,22,23]. In order to implement quality improvement projects in a collaborative and technically sound manner, we have developed the St. Jude Global Packages. These “packages” are designed to develop solutions for intervening and improving care locally and informing implementation of these endeavors globally [24]. The St. Jude Global Packages have been conceptualized as practice improvement interventions with cumulative multi-site experience and a measurement strategy, and collaboratively designed with global partners with the intent of global dissemination.

This specific St. Jude Global Package draws from the success of the “Children’s Comfort Promise” in high-resource settings to collaboratively develop with our partners in LMICs a QI intervention for pain during needle-based procedures that is acceptable for implementation in a range of resource and cultural settings around the world [13]. Between July 2021 and January 2023, a pilot of the “Global Comfort Promise” was performed in four St. Jude Global Alliance Partner sites around the world (Instituto Nacional de Enfermedades Neoplásicas (INEN) in Lima, Peru, Hospital de Amor in Barretos, Brazil, Grey’s Hospital in Pietermaritzburg, South Africa, and Philippines General Hospital in Manila, Philippines). These sites were selected as they had pediatric palliative care champions at the institution who had previously successfully worked with St. Jude Global on other improvement initiatives and had identified the management of pediatric pain as a priority at their institution.

Instituto Nacional de Enfermedades Neoplásicas is a cancer hospital in Lima, Peru, and cares for both adults and children. It is the largest pediatric cancer facility in the country and cares for over 800 new pediatric cancer diagnoses per year. Hospital de Amor in Barretos, Brazil, is a stand-alone pediatric cancer hospital and the largest independent pediatric cancer hospital in Latin America. It cares for over 350 new cases of cancer each year. Grey’s Hospital in Pietermaritzburg, South Africa, is a tertiary referral center that serves Kwazulu-Natal Province. It has a dedicated pediatric oncology unit. Philippines General Hospital in Manila, Philippines, is a tertiary referral hospital with a 1500 bed capacity and has around 300 new cases of pediatric oncology per year.

### 2.2. Interventions

A 4-part (4 sessions over 2 weeks) workshop was conducted in July 2021 to co-design the QI intervention with the international sites along with providing education on QI theory and methods [25]. This workshop was attended by an expert panel of diverse multi-disciplinary, geographical, and resource-level representation, including participation from each of the institutions where the Global Comfort Promise intervention was implemented. Figure 1 shows the structure of the 4-part workshop. During this time, the package was presented to institution teams along with training in improvement science methodology (See Appendix A). By the end of the workshop, the group identified a common aim to accomplish, developed measures to assess improvement, created a driver diagram and change ideas (Figure 2), and identified components of the change package that could be adapted and the elements that required fidelity across sites.

Additionally, a SMART (Specific, Measurable, Achievable, Realistic, Timely) project aim was created for the package, and each site developed their own site-specific aim that was consistent with their cultural context, resources, and capacity to achieve the overall global change package aim. The SMART aim for the Global Comfort Promise was to increase by ≥60% the utilization of interventions to prevent or alleviate procedural pain from needlesticks performed in children with cancer by applying the principles of the Global Comfort Promise at St. Jude Global partner institutions in the first 6 months of implementation. SQUIRE 2.0 Guidelines were followed in the development and implementation of the intervention along with reporting of the findings and results [20].

The decision on where to implement the Global Comfort Promise was based on the specific needs, capability, and capacity of the interdisciplinary team and infrastructure at each site. The team in the Philippines instituted the Global Comfort Promise in their outpatient clinic, while the team in South Africa instituted the Global Comfort Promise in the inpatient unit. The team in Brazil decided to implement the project in both the inpatient and outpatient setting, while the team in Peru started in the inpatient setting before transitioning to the outpatient setting when they realized that most patients had PIVs placed or central lines accessed prior to admission.

### 2.3. Measures

The data collection forms collected data to assess outcome, process, and balance measures to assess the impact of the Global Comfort Promise on patient experience. Appendix A shows the outcome, process, and balance measures, including their operational definition and the forms in which this information was obtained. Outcome measures specifically assessed patient/parent satisfaction with the prevention and alleviation of pain after a needlestick. As this information was not routinely collected prior to participation in this initiative, there are no baseline data available. Process measures included the frequency of use of each Comfort Promise intervention and the availability of needed materials in the clinical area, including topical anesthesia and distraction supplies. Prior to participation in this QI initiative, none of the sites consistently applied the principles of the Comfort Promise bundle and so the baseline was considered to be zero for each process measure. Due to concerns about consistent access to topical anesthesia and a small number of infants in the patient population, the core process measure of adherence was the use of at least two Comfort Promise principles. Additionally, due to variability in the types of topical anesthesia available in the different countries, either 4% lidocaine cream (LMX) or a eutectic mixture of local anesthetics (EMLA) was acceptable for use. Balance measures included time spent waiting for the procedure along with time the procedure required and whether the procedure was successful or not.

### 2.4. Study of the Intervention

To assess the impact of the change ideas and interventions on patient experience, 3 standardized data collection forms (See Appendix A) were developed. These data collection forms assessed the experience directly from three primary respondents (the patient or their parent, the provider performing the procedure, and the implementation team member conducting the audits). Data were collected on paper first, then transferred to an electronic QI platform (SimpleQI) case-by-case [26]. The data collection strategy was based on sampling and the forms did not contain patient identifiers, maintaining anonymity. The individual collecting the data collection forms varied between institutions based on the availability of human resources.

The data collection forms were developed according to the Model for Improvement methodology to capture the data according to the measures co-developed during the second session of our workshop, which assessed the question “How will we know that a change is an improvement?” The patient/parent report form was developed to assess the outcome measures, including whether the patient experienced pain, which was a simple yes/no option. This was based off a validated patient satisfaction questionnaire [13,27]. Additionally, the patient/parent was asked about their satisfaction; this was a simple Likert scale based on Likert scale development in LMICs [28]. The last 2 questions assessed the location of the procedure and wait time for the procedure as balance measures. The provider self-report form included the process measures, including self-assessment to ensure that each of the different interventions of the bundle were performed, and the balance measures, including the success of the procedure, number of attempts, and the healthcare professional’s satisfaction in performing the procedure. Finally, the auditing form included the questions from both the patient/parent and self-report forms. This form was completed one time per week on different days and at different times to avoid bias and to provide an independent third-party observation. In addition to the questions found on the patient/parent and self-report forms, the auditing form had free text space to obtain open-ended feedback from the patients/parents and healthcare professionals.

Data were collected at least one day per week for each form for a minimum of 5 needlesticks and a maximum of 20 needlesticks. The specific day of the week, time of day, and clinical setting when and where data collection occurred varied and fluctuated to capture the relevant clinical settings and circumstances as these elements were captured in the data collection form. The data collection forms were translated to local languages (Spanish, Portuguese, Tagalog, and Zulu) for patient and parent completion.

### 2.5. Analysis

Data analysis included the observation of trends in institutional run charts, as small multiples and overall, for a temporally related cohort of institutions and at package evaluation timepoints. Binary measures were analyzed as monthly frequencies and continuous data as monthly means. Run charts were stratified by relevant variables such as day of the week, clinical area, or patient age (in years). Control charts were generated once >20 data points are available. Standard run chart and control chart rules were applied to adjust or recalculate the median or mean, respectively, and to assess for common and special cause variation [29,30].

### 2.6. Ethical Considerations/IRB Approval

The Office of Human Subjects Research Protections and Institutional Review Board at St. Jude Children’s Research Hospital (Memphis, TN, USA) approved this study as exempt research. Additional local approvals were obtained as needed.

## 3. Results

### 3.1. Demographics

A total of 2,185 individual procedures were recorded in the provider self-report forms and were performed from January 2022 to January 2023 at the four institutions. Most patients were male (56.7%), with 1.7% of the patients 12 months or younger, 58.9% under the age of 10, and 39.4% 10 to 18 years old (Table 1). The most common diagnoses were solid tumors (37.9%), followed by acute lymphoblastic leukemia (35.1%) and central nervous system tumors (9.7%). Over half of the needle-based procedures were lab draws (55.9%), while the rest were peripheral IV placements or central line access (44.1%).

Most procedures were performed by nurses (59.5%), while some procedures in Brazil were performed by nursing technicians (31%). In South Africa, the procedures were performed by physician registrars or interns (Table 1).

### 3.2. Outcome Measures

A total of 2199 patient/parent satisfaction forms were also collected. Patients and parents expressed high satisfaction right after needlestick procedures (Figure 3A), with a median of 95.7% satisfaction throughout the time studied. When asked whether the patient was in pain during the procedure, over the time studied, the number of patients decreased from a high of 61.7% at the beginning of this study to a median of 32.1% (Appendix A). When looking at those patients who did report pain during the procedure, 94.7% reported that the team “definitely” did everything in their power to address the pain, while the other 5.3% said yes, to some extent. No patients or parents felt like the team did not attempt to address their pain (Appendix A).

### 3.3. Process Measures

In the time studied, all four sites were successful in implementing at least two Comfort Promise principles, with a median of 97.8% for the time evaluated (Figure 3B). Distraction techniques and comfort positioning were the most consistently applied principles. The median percent use of distraction from January 2022 to January 23 was 96.9% (Figure 3C). Each site implemented unique distraction techniques based on resources. Common distraction tools included music, bubbles, pinwheels, action figures and dolls, and toys with flashing lights, with the most common tool being a smartphone if available for adolescents. The team in Peru was able to utilize virtual reality headsets donated to the hospital as an additional mechanism of distraction for older children and adolescents. The median percentage of comfort positioning techniques was also high, at 99.7% (Figure 3D).

A lesson learned when evaluating the processes of needle-based procedures was that many children would have multiple needle pokes in one day. For example, the team in the Philippines identified that when a child arrived, they would have their labs drawn and if their labs were sufficient for chemotherapy, they would have a subsequent needle-based procedure to place the PIV to initiate treatment (Appendix A). This resulted in patient distress due to multiple pokes during each visit. The process was subsequently changed to prioritize one needle-based procedure whenever possible.

The use of sucrose or breastfeeding was less commonly applied as compared to distraction and comfort positioning techniques, with a median of 36.4% (Appendix A). January 2023 through August 2023 had a higher median of 69.1% but over the subsequent months, it trended down to no use in the month of December 2023 before recovering a bit in January 2023 after reeducation at the sites. Additionally, there were very small numbers of children who qualified for this measure, as most patients were two years or older. This resulted in significant variability in the percentage receiving this intervention due to the small denominators (Appendix A). Several factors impacted the use of this intervention. First, in the first 6 months of implementation, the operational definition was for breastfeeding or sucrose in infants less than 12 months of age, to be consistent with what was done in the United States. However, in discussion with the implementation teams at each site, they encouraged broadening the definition to include children up to 24 months and even up to 36 months, as this was more consistent with cultural practices. This increased the number of patients to whom breastfeeding was to be offered. Second, many parents and family members declined breastfeeding as they felt it would take longer.

Additionally, topical anesthesia was not used as frequently as comfort positioning or distraction. The median use over the timeframe assessed was 39.6% (Appendix A). Like breastfeeding and sucrose, the use of topical anesthesia was best in the first half of the year and trailed off in the second half of the year. In discussion with the teams, there were several explanations for this drop off. First, there were concerns about inconsistent access to topical anesthesia due to supply chain issues, not having it on formulary at their institutions, and intermittent stockouts. However, by the end of the year, all sites regularly had access to topical anesthesia. The second and larger concern was that because the core process measure focused on at least two interventions being consistently implemented in each needlestick procedure, the staff performing the procedure would often bypass breastfeeding and topical anesthesia and focus more on the distraction and positioning techniques. Despite this, there were some successes as families and patients began consistently asking for topical anesthesia as they saw the impact. Several patients in the Philippines referred to the topical anesthesia as “magic lotion” and the median use in the Philippines was 91.5% during the period assessed. The total number of interventions for each reported encounter broken down by age can be found in Appendix A.

### 3.4. Balance Measures

Throughout the time studied, there was not a pronounced difference in the amount of time required to perform the needlestick procedure, with a median of 5.6 min required for the procedures. There was a slight increase in the first few months to a maximum of 8.5 min per procedure before steadily decreasing to 3.1 min by January 2023 (Appendix A). When discussing this with the implementation sites, they felt this was due to the providers and parents becoming accustomed to the principles of the Global Comfort Promise and as they assimilated to the process, it became easier. Additionally, the median success rate for the procedures was 100% throughout the time studied, with the median number of attempts being 1.24 (Appendix A).

Overall, the healthcare professionals were satisfied with the process, with a median satisfaction of 99.1% (Figure 4, Appendix A). When debriefing with the healthcare professionals, they felt that once they became used to the process of the interventions to address pain during needle-based procedures, they did not feel like the additional steps were more burdensome or greatly impacted their workflow. In fact, several described how it simplified their process, as the patients were less combative or anxious.

Not only did the healthcare professionals describe how the intervention impacted their workflow, but they also discussed its impact on their experience as well. In the auditing form, there were several open-ended responses on the experience of the healthcare professionals and how it impacted them. For example, one physician in South Africa described that due to their experience with the Global Comfort Promise, “*I started paying attention to ways I could improve the kids’ experiences during procedures…I felt less anxious about doing painful procedures and I think this made a difference to my success rate and confidence.*”

Another stated, “*The study highlighted for me the value of being a kind and compassionate doctor. Just by holding and soothing a child during a procedure or getting the mom to sit nearby and hold their hand, a very scary and potentially traumatic event could be avoided*.”

## 4. Discussion

Pain is common, underrecognized, and undertreated, with the most common source of pain being needle-based procedures [1,2,3,4]. For children with cancer in LMICs, this burden is exacerbated due to the frequent needle pokes a patient will require for their treatment. This report demonstrates the feasibility of introducing principles to reduce or eliminate pain from needle-based procedures (Global Comfort Promise) in unique clinical settings around the world, resulting in high patient and parent satisfaction without increasing procedure time or an adverse effect on provider satisfaction.

Each site was successful in their own adoption, adaptation, and implementation of the bundle while navigating a host of challenges. In South Africa, there were devastating floods, along with public sector worker strikes in 2022 that impacted patient care and the supply chain for supplies such as topical anesthesia [31,32]. In Peru, the president of the country was arrested after attempting to dissolve congress, leading to months of protests and political violence which disrupted the public healthcare system as a whole [33,34]. Additionally, the implementation of this project continued in the shadow of the COVID-19 pandemic and the restrictions placed on healthcare professionals throughout all the sites involved frequent turnover as nurses, physicians, and other healthcare professionals navigated an uncertain healthcare environment. Despite these challenges, each site was able to adapt and adopt the Global Comfort Promise bundle in their own way and ensure that there was a continued momentum and interest in addressing pain during needle-based procedures despite limited resources, navigating a global pandemic, and instability from political conflict or climate change.

One unexpected consequence of the implementation of this bundle was the importance of cultural adaptation and the need for further refinement of the bundle. For example, breastfeeding was a common practice in the community but went beyond the 12 months that were identified in the US-based cohort. All four sites highlighted that culturally, breastfeeding is common up to 2 years of age and even beyond. This is consistent with WHO recommendations, especially in LMICs, to ensure infants and toddlers have access to more beneficial sources of nutrition [35]. For this reason, the change to include children up to 2 years of age was made for this measure. However, this did lead to the unintended consequence of having the overall implementation of the intervention decrease, as there were more potential patients to include without an adequate uptick in the use of the intervention.

Another unexpected consequence was the recognition that the bundle had four interventions for children less than 2 years of age but for older patients, there were only three interventions: positioning, distraction, and topical anesthesia. For this reason, many sites incorporated other elements of support for the patients, including praise and rewards after the needle-based procedure as a part of their distraction, especially for those children older than 2 years of age. There is existing literature on the importance of praise and reward after needle-based interventions for pediatric patients. For future implementation of the Global Comfort Promise bundle, praise and reward after a needle-based procedure will be incorporated as an additional step for children older than 2 years of age [36,37,38].

Offering topical anesthesia to children undergoing needle procedures is an integral part of preventing the pain and anxiety caused by needles and must not be omitted or substituted. Both in high-income countries as well as in LMICs, the resistance of staff is high. One challenge was that, due to concerns about consistent access to topical anesthesia, one of the key process measures for success was having two or more elements of the bundle implemented during each needlestick, rather than all the modalities spelled out by the original Comfort Promise QI protocol. This resulted in success with a median of 97.8% of procedures including at least 2 elements of the Comfort Promise bundle. However, when all four sites eventually obtained more secure access to topical anesthesia, the overall use of topical anesthesia actually went down during the time studied. This is an example of a principle known as Goodhart’s law, named after the famed British economist and phrased by the anthropologist Marilyn Strathern [39,40]. Goodhart’s law states that “When a measure becomes a target, it ceases to be a good measure”. There is concern that focusing on the measure of at least two elements of the Global Comfort Promise and the high satisfaction of parents (which research has shown does correlate with pain intensity) [41] discouraged sites from implementing all elements of the bundle. Each site is currently working on implementing topical anesthesia for all needle-based procedures in the second year of implementation and for future sites participating in the Global Comfort Promise QI initiative, we will remove the measure of at least two components of the bundle. For future sites participating in this QI initiative, topical anesthesia and sucrose/breastfeeding for infants and babies will become an integral part of the QI project, with the authors supporting overcoming conceptual, perceived, and real barriers to administering topical anesthesia.

### Limitations

This project had several limitations. First, as mentioned, this was a multi-institutional pilot with hospitals in different resource and cultural contexts, which may have impacted the findings. However, we worked with all the sites to ensure fidelity within the core measures and allow for necessary adaptation in implementation. Additionally, while we demonstrated high satisfaction from both patients and families along with the healthcare professionals, these were obtained through simple scales and may not reflect more nuanced views of the experience. In future work, we hope to have some more qualitative and open-ended responses to better understand the experience of the patients/families as well as the medical team. With this, patient/parent satisfaction was consistently high throughout implementation, while the reporting of pain had a noticeable decrease over time. This may be a result of patients/parents not being willing to share their dissatisfaction with care due to different cultural contexts and for future work, the presence of pain will be a core outcome measure to better assess patient experience. Another limitation is a lack of data prior to participation in this project. Therefore, for these outcome measures specifically, we do not know what previous levels of satisfaction had been for patients and parents for comparative purposes. Finally, the sites implemented this initiative in different hospital units, as well as each institution having different healthcare professionals who performed the needle-based procedures. This may have impacted generalizability. Over the coming year, each site is planning for expansion and scalability to other units within the hospital to ensure sustainability in their institutions.

## 5. Conclusions

This manuscript highlights the feasibility of a global multi-site QI initiative (Global Comfort Promise) to address pain during needle-based procedures for children with cancer in LMICs despite unique challenges at each site. There was high satisfaction amongst the patients and caregivers along with the healthcare team in the implementation of this project. Further work is needed to ensure sustainability and scalability at the pilot sites along with some refinement of the measures to ensure all elements of the bundle are implemented. The findings of this project will guide the further development of the package at each site as well as expansion to other sites in resource-constrained settings.

## Figures and Tables

**Figure 1 cancers-16-01025-f001:**
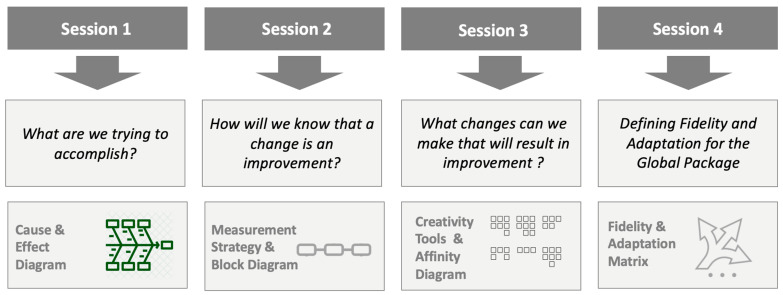
Collaborative workshop. A 4-part workshop was conducted in July 2021 in which each content expert from around the world developed the aim, measurement strategy, and driver diagram for the Global Comfort Promise QI initiative. The final session identified principles of fidelity and adaptation to allow for consistency between the sites but also to allow for flexibility based on institutional resources and needs.

**Figure 2 cancers-16-01025-f002:**
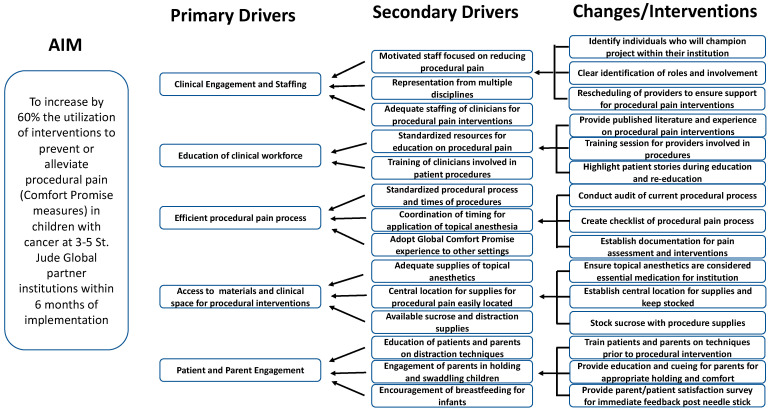
Global Comfort Promise driver diagram developed by Global Comfort Promise expert team. This includes the primary drivers of clinical engagement and staffing, education of the clinical workforce, efficient procedural pain processes, access to materials and clinical space for the procedures to occur, and finally patient and parent engagement. Secondary drivers and change ideas were identified for each primary driver.

**Figure 3 cancers-16-01025-f003:**
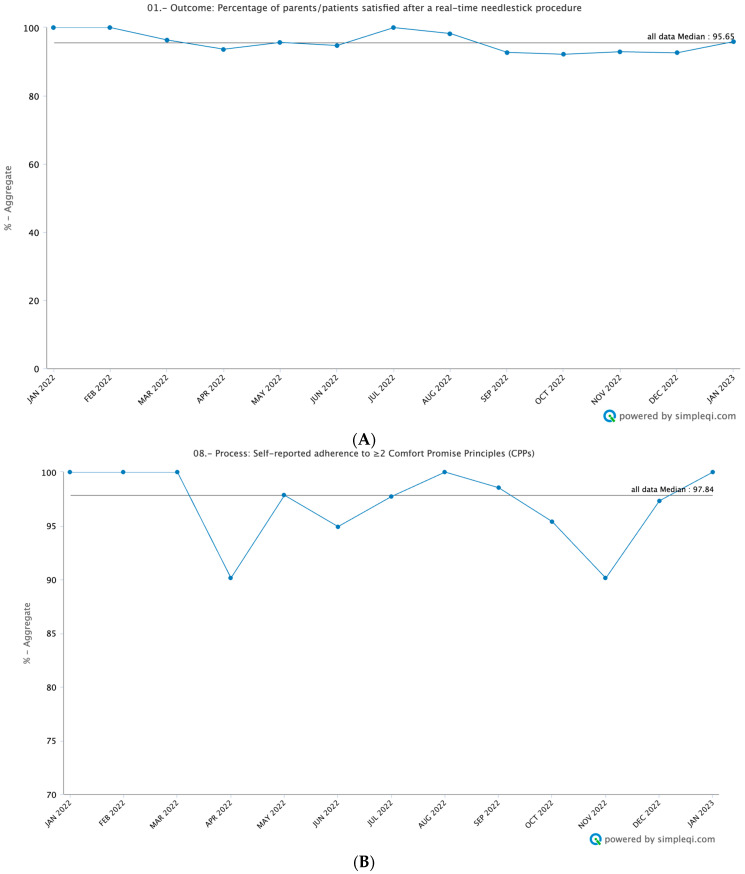
(**A**). Patient/parent satisfaction after real-time needlestick procedure. Percentage of patients/parents that stated that the patient did not have any pain during the procedure or if they did experience pain, they stated that “yes, definitely” the healthcare team did everything to address their pain over the total number of procedures recorded. These data were collected each week and then aggregated into monthly data points for analysis. (**B**). Self-reported adherence to ≥2 Comfort Promise principles. Percentage of healthcare professionals that recorded using at least 2 Comfort Promise interventions during the needle-based procedures performed over the total number of needle-based procedures performed. These data were collected each week and then aggregated into monthly data points for analysis. (**C**). Self-reported percentage use of age-appropriate distraction Techniques. Percentage of healthcare professionals that recorded using age-appropriate distraction techniques during the needle-based procedures performed over the total number of needle-based procedures performed. These data were collected each week and then aggregated into monthly data points for analysis. (**D**). Self-reported percentage of age-appropriate positioning techniques. Percentage of healthcare professionals that recorded using age-appropriate positioning techniques during the needle-based procedures performed over the total number of needle-based procedures performed. These data were collected each week and then aggregated into monthly data points for analysis.

**Figure 4 cancers-16-01025-f004:**
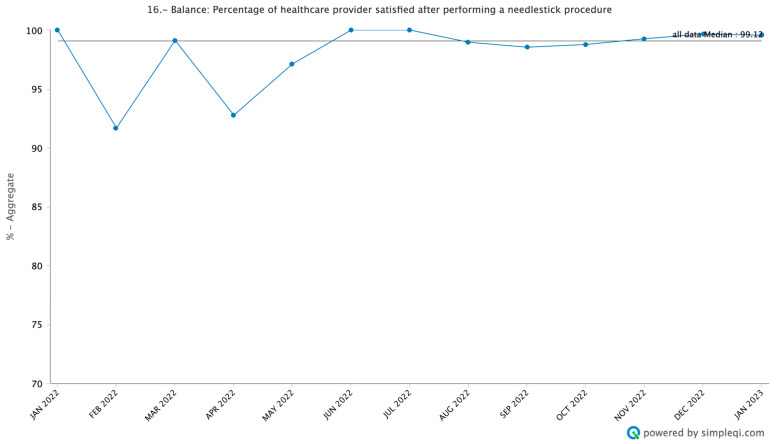
Provider satisfaction after needlestick procedure. The percentage of healthcare providers that reported they were very satisfied or satisfied with the procedure after completion of the needlestick. These data were collected each week and then aggregated into monthly data points for analysis.

**Table 1 cancers-16-01025-t001:** Demographics.

Patient	2185
**Provider Type**	
Nurse	1303 (59.6%)
Nursing technician	678 (31.0%)
Physician	204 (9.4%)
**Gender**	
Female	944 (43.3%)
Male	1241 (56.7%)
**Age**	
0–12 months	37 (1.7%)
1–3 years	386 (17.6%)
4–6 years	507 (23.2%)
7–9 years	395 (18.0%)
10–12 years	295 (13.5%)
13–15 years	315 (14.4%)
16–18 years	250 (11.4%)
**Diagnosis**	
Acute lymphoblastic leukemia	769 (35.1%)
Acute myeloblastic leukemia	104 (4.8%)
Central nervous system tumors	212 (9.7%)
Hodgkin’s lymphoma	120 (5.5%)
Non-Hodgkin’s lymphoma	49 (2.2%)
Solid tumor	829 (37.9%)
Other * JMML	102 (4.7%)
**Type of Needlestick**	
Lab draw	1223 (55.9%)
Peripheral IV placement	962 (44.1%)

* JMML: juvenile myelomonocytic leukemia.

## Data Availability

Data available upon request.

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
