# Peer review of "Preventing and Treating Pain and Anxiety during Needle-Based Procedures in Children with Cancer in Low- and Middle-Income Countries"

_cancers, 2024, doi:10.3390/cancers16051025_

Round 1
Reviewer 1 Report
Comments and Suggestions for Authors
What was the baseline? I.e. what was the practice at the 4 sites before implementing the programme?
What were the outcome measures before the programme?
What were the specific distraction interventions?
What was specifically the local anestehsia?
Why (supplement) is there a huge variance over months for measures presented in Gigure S6? Sometimes form 0 to 100%?
Author Response
Reviewer 1:
What was the baseline? I.e. what was the practice at the 4 sites before implementing the programme?
This is an important question. None of the sites were performing any of the Comfort Promise principles consistently and so the baseline was zero for the different components.
We have added this to the manuscript under the Methods section entitled 2.4 Measures we state “Prior to participation in this QI initiative none of the sites consistently applied the principles of the Comfort Promise bundle and so the baseline was considered to be zero for each process measure.”
What were the outcome measures before the programme?
Thank you so much for this question. These were not collected prior to participation in this QI initiative and therefore we do not have a baseline.
We have included this in the Methods section entitled 2.4 Measures where we state “As this information was not routinely collected prior to participation in this initiative, there is no baseline data available” Additionally, in the Limitations section we say the following “Another limitation is a lack of data from prior to participation in this project. Therefore, for the outcome measures specifically we do not know what the satisfaction was for patients and parents for comparative purposes.”
What were the specific distraction interventions?
We appreciate this question. Each site had different distraction interventions based on resources and availability. These included bubbles, pinwheels, music, action figures, toys with flashing lights, and often smartphones for adolescents, and even virtual reality headsets in Peru.
We have adjusted the Results section under Process Measures to describe the following. “Each site implemented unique distraction techniques based on resources. Common distraction tools included music, bubbles, pinwheels, action figures and dolls, toys with flashing lights, with the most common tool being a smartphone if available for adolescents. The team in Peru was able to utilize virtual reality headsets donated to the hospital as an additional mechanism of distraction for older children and adolescents.”
What was specifically the local anesthesia?
Thank you for this question. The topical anesthesia was 4% licodaine cream (LMX) or EMLA, based on what was available at the institution knowing there is variability in country and institutional access.
We have added the following in the Methods section under Measures, “Additionally, due to variability in types of topical anesthesia available in the different countries, either 4% lidocaine cream (LMX) or eutectic mixture of local anesthetics (EMLA) was acceptable to use.”
Why (supplement) is there a huge variance over months for measures presented in Figure S6? Sometimes from 0 to 100%?
This is an important point. The variance is because of the very small numbers of children in the age range for breastfeeding. Because of the small denominator this resulted in a wide variance as even having 1 child breastfeeding out of 2 possible patients would be 50% versus 0% of 2/2.
We have added the following in the results section under Process measures which reads as follows “Additionally, there were very small numbers of children which qualified for this measure as most patients were 2 years or older. This resulted in significant variability in the percentage receiving this intervention due to the small denominators (Supplemental Figure 6A).”
Reviewer 2 Report
Comments and Suggestions for Authors
The authors of the manuscript entitled ‘Preventing and Treating Needle-Based Procedural Pain and Anxiety for Children with Cancer in Low- and Middle-Income Countries’ present an interesting study performed on a relevant population. Besides the fact that procedural pain during childhood is a known risk factor for increased pain sensitivity and health care avoidance in general, they very convincingly substantiate: ‘…many health systems in LMICs do not have capacity for central venous lines resulting in children with cancer requiring dozens to even hundreds of needle pokes during their treatment course. The avoidance of repeated painful procedures has been identified as a cause of early treatment abandonment and mortality in LMICs’. The manuscript is well written, the design and preparation seems well considered and supported by experts, although there are some methodological flaws, as acknowledged in the limitation section by the authors. The manuscript is relevant because it encourages improvement of care, based on partly low/no cost interventions such as breast feeding and distraction. See below for some final remarks.
Methods
Page 3 line 128 - page 5 line 160 The interventions are well elaborated in workshops, reflected by figure 1 and 2. However, the lowest row of figure 1 is unreadable. Is it meant as illustration, or otherwise please provide a clearer figure in the supplemental file?
Page 5 line 163 …3 standardized data collection forms were developed. This seems the major flaw of the study, since obviously no validated scales were used? Although this was acknowledged in the limitation section, I would like the authors to reflect more on the development of the data collection forms. E.g., were they derived from other questionnaires, how and by whom was the selection of items and number of answering options performed?
Results
Table 1 Please remove ‘5’ (typo?) before Patient. ‘Other:’ before Nursing Technician seems redundant, please remove. Please explain *JMML in legend.
Page 6 line 220 Please explain PIV when used the first time.
Page 7 line 248 Supplemental figure 5 seems to belong to page 9 line 269. Please correct.
Results general:
1 Although the total numbers of collected data forms are reported, only for table 1 and figure 3A specific numbers are reported in the results or supplemental file. For the remaining outcomes only percentages are presented. It would be informative to have a table with the actual numbers concerning the number of procedures that were performed using only one, two, three or all four measures (split for cut-off age 2 years because of sucrose/breastfeeding). Similarly for the scores on satisfaction of the provider: could the authors provide a table with the numbers (%) showing the whole range from not satisfied to very satisfied, in the supplemental file? The totals over the whole study period -aggregated year number- would be sufficient.
2 The supplemental file provides an audit form, figure 3. Can the authors reflect on the use of this form, who audited and how often, and the data collected by it?
Discussion
Pain scores are high during the first months, however, the satisfaction of parents/patients seems high from the first month. The satisfaction of health care providers seems more in line with the higher pain scores in the beginning. I would like the authors to reflect on this.
Author Response
Reviewer 2:
The authors of the manuscript entitled ‘Preventing and Treating Needle-Based Procedural Pain and Anxiety for Children with Cancer in Low- and Middle-Income Countries’ present an interesting study performed on a relevant population. Besides the fact that procedural pain during childhood is a known risk factor for increased pain sensitivity and health care avoidance in general, they very convincingly substantiate: ‘…many health systems in LMICs do not have capacity for central venous lines resulting in children with cancer requiring dozens to even hundreds of needle pokes during their treatment course. The avoidance of repeated painful procedures has been identified as a cause of early treatment abandonment and mortality in LMICs’. The manuscript is well written, the design and preparation seems well considered and supported by experts, although there are some methodological flaws, as acknowledged in the limitation section by the authors. The manuscript is relevant because it encourages improvement of care, based on partly low/no cost interventions such as breast feeding and distraction. See below for some final remarks.
We appreciate Reviewer 2 for taking the time to review our article and provide suggestions and feedback. We are hopeful our experience can help others address needle-based procedural pain in resource-constrained settings around the world. Thank you for the positive feedback and thoughtful suggestions.
Methods
Page 3 line 128 - page 5 line 160 The interventions are well elaborated in workshops, reflected by figure 1 and 2. However, the lowest row of figure 1 is unreadable. Is it meant as illustration, or otherwise please provide a clearer figure in the supplemental file?
We appreciate this feedback. The original figure was larger and appears to have been edited to fit within the manuscript file. It is meant as an illustration, and to avoid confusion we have removed the lowest row to simplify the figure.
Page 5 line 163 …3 standardized data collection forms were developed. This seems the major flaw of the study, since obviously no validated scales were used? Although this was acknowledged in the limitation section, I would like the authors to reflect more on the development of the data collection forms. E.g., were they derived from other questionnaires, how and by whom was the selection of items and number of answering options performed?
Thank you for this important suggestion. The data collection forms were developed according to the Model for Improvement Methodology to capture the data according to the measures co-developed during the second session of our workshop assessing “How will we know that a change is an improvement?” We have rearranged the section in the Methods sections on “Measures” before the section “Study of the Intervention” to provide context on the measures before describing the forms. Additionally, we have included the following paragraph in the “Study of the Intervention” section to provide more context on the methodology behind the data collection forms.
“The data collection forms were developed according to the Model for Improvement Methodology to capture the data according to the measures co-developed during the second session of our workshop assessing “How will we know that a change is an improvement?” The patient/parent report form was developed to assess the outcome measures including whether the patient experienced pain which was a simple yes/no option. This was based off a validated patient satisfaction questionnaire. [13,28] Additionally, the patient/parent was asked regarding satisfaction and was a simple Likert scale based on Likert scale development in LMIC.[29] The last 2 questions assess location of the procedure and wait time for the procedure as balancing measures. The provider self-report form included the process measures including self-assessment to ensure that each of the different interventions of the bundle were performed, and the balance measures including the success of the procedure, number of attempts, and the healthcare professional satisfaction in performing the procedure. Finally, the auditing form included the questions from both the patient/parent, and self-report form was completed one time per week on different days and time to avoid bias and to provide a third-party independent observation. In addition to the questions found on the patient/parent and self-report form, the auditing form had free text space to get open-ended feedback from the patients/parents and healthcare professionals.”
Results
Table 1 Please remove ‘5’ (typo?) before Patient. ‘Other:’ before Nursing Technician seems redundant, please remove. Please explain *JMML in legend.
We appreciate these suggestions. They have been changed in Table 1.
Page 6 line 220 Please explain PIV when used the first time.
Thank you for this suggestion. In looking back at the manuscript we refer to peripheral intravenous lines in the second paragraph of the introduction but abbreviate it to IV. We have changed this to PIV to be consistent throughout.
Page 7 line 248 Supplemental figure 5 seems to belong to page 9 line 269. Please correct.
Thank you for catching this. We have removed the erroneous reference to Supplemental Figure 5 and updated it to refer to the experience of the team in the Philippines.
Results general:
1 Although the total numbers of collected data forms are reported, only for table 1 and figure 3A specific numbers are reported in the results or supplemental file. For the remaining outcomes only percentages are presented. It would be informative to have a table with the actual numbers concerning the number of procedures that were performed using only one, two, three or all four measures (split for cut-off age 2 years because of sucrose/breastfeeding). Similarly for the scores on satisfaction of the provider: could the authors provide a table with the numbers (%) showing the whole range from not satisfied to very satisfied, in the supplemental file? The totals over the whole study period -aggregated year number- would be sufficient.
We appreciate this feedback. We have now included the following information in the Supplemental Materials in Supplemental Table 3: Number of Comfort Promise Interventions by Age, and Supplemental Table 4: Provider Satisfaction. We have also made the following references in the text,
“The total number of interventions for each reported encounter broken down by age can be found in Supplemental Table 3.” And “Overall, the healthcare professionals were satisfied with the process with a median satisfaction of 99.1% (Figure 4, Supplemental Table 4).
2 The supplemental file provides an audit form, figure 3. Can the authors reflect on the use of this form, who audited and how often, and the data collected by it?
Thank you for this suggestion. We have included this in the Measures sections under the subheading “Study of the Interventions.” We have no added the following “Finally, the auditing form included the questions from both the patient/parent, and self-report form to provide a third-party independent observer once per week at different days and times to verify the results of the patient/parent and self-report forms as well as have open-ended text space to get open-ended feedback from the patients/parents and healthcare professionals.”
Discussion
Pain scores are high during the first months, however, the satisfaction of parents/patients seems high from the first month. The satisfaction of health care providers seems more in line with the higher pain scores in the beginning. I would like the authors to reflect on this.
This is an important observation, and we appreciate Reviewer 2 highlighting this. We have added the following to the Limitations section in the Discussion, “Additionally, while we demonstrated high satisfaction from both patients and families along with the healthcare professionals, these were obtained through simple scales and may reflect more nuanced views of the experience. Future work we hope to have some more qualitative and open-ended responses to better understand the experience of the patient/family as well as the medical team. With this, the patient/parent satisfaction was consistently high throughout implementation while the reporting of pain had a noticeable decrease over time. This may be a result of patients/parents not willing to share their dissatisfaction with care due to different cultural contexts and for future work, presence of pain will be a core outcome measure to better assess the patient experience.”
Reviewer 3 Report
Comments and Suggestions for Authors
Dear Authors, thank you for allowing me to review this interesting manuscript describing the feasibility of the implementation of an intervention to reduce pain in children with cancer. I found the topic timely and of interest to the readership of the journal.
Although the manuscript is well written and fluent, I have some concerns that should be addressed before it is recommended for acceptance. I list some suggestions below that could contribute to the improvement of the overall quality of your presentation.
INTRODUCTION
Page 2, lines 67-75. Please, improve the consistency of your terms; the interventions are labelled as system-wide and process, the risk for the reader is to be confused by the terms you have used.
METHODS
Page 3, lines 120-127. Please, I would like some contextual information on the centres in which the interventions were implemented to be presented more in depth. There is some information that could be placed here and put in the results section, see my comment below.
Page 5, line 165. You have used the term audits here, whereas before you have labelled the interventions as "workshops".
RESULTS
All along the section, medians are presented as percentages, is it correct?
Page 6, Lines 214-223. This information could be used in the methods section to present the contexts in which the interventions were implemented.
Page 7, section 3.3. Is it possible to have a table in which comfort principles, distraction techniques, and positioning techniques are described in detail?
Pages 10-11, Lines 320-328. How do these qualitative parts were collected and analysed? This was not described in the methods section.
DISCUSSION
Page 11, From line 362. Is the sample included generalizable? The low number of infants 0-4 years could have affected the practices you have described? Moreover, a discussion about the variety of professionals performing the tasks and the potential impact on the presented results is not addressed.
I hope these suggestions will help you
Best regards
Author Response
Reviewer 3:
Dear Authors, thank you for allowing me to review this interesting manuscript describing the feasibility of the implementation of an intervention to reduce pain in children with cancer. I found the topic timely and of interest to the readership of the journal.
Although the manuscript is well written and fluent, I have some concerns that should be addressed before it is recommended for acceptance. I list some suggestions below that could contribute to the improvement of the overall quality of your presentation.
We appreciate Reviewer 3 for taking the time to review our manuscript and are grateful for their feedback and suggestions to improve.
INTRODUCTION
Page 2, lines 67-75. Please, improve the consistency of your terms; the interventions are labelled as system-wide and process, the risk for the reader is to be confused by the terms you have used.
We appreciate this suggestion. We have attempted to use consistent terminology with “intervention” instead of “process” or “protocol”. We have also removed extraneous use of the term system-wide except for when explicitly describing how the intervention was scaled up “system-wide” across the hospital system.
METHODS
Page 3, lines 120-127. Please, I would like some contextual information on the centres in which the interventions were implemented to be presented more in depth. There is some information that could be placed here and put in the results section, see my comment below.
Thank you for this suggestion. We have moved the information that was in the Methods section into the Results section into the Methods section to provide context to the implementation of this QI intervention at the different hospitals. We also, provided a brief sentence about each institution including location, hospital size, and resources.
The following is now found in the Methods section under Context we have added the following paragraph, “Instituto Nacional de Enfermedades Neoplásicas is a cancer hospital in Lima, Peru and cares for both adults and children. It is the largest pediatric cancer facility in the country and cares for over 800 new pediatric cancer diagnoses per year. Hospital de Amor in Barretos, Brazil is a stand-alone pediatric cancer hospital and the largest independent pediatric cancer hospital in Latin America. It cares for over 350 new cases of cancer each year. Grey’s Hospital in Pietermaritzburg, South Africa is a tertiary referral center that serves Kwazulu-Natal Province. It has a dedicated pediatric oncology unit. Philippines General Hospital in Manila, Philippines is a tertiary referral hospital with a 1,500 bed capacity and has around 300 new cases of pediatric oncology per year.”
Page 5, line 165. You have used the term audits here, whereas before you have labelled the interventions as "workshops".
We appreciate this reviewer’s clarification. The workshop was conducted over 4 sessions prior to the implementation of the intervention to develop the Global Comfort Promise in collaboration with the 4 participating institutions. The goal of this workshop was to design the strategy including what to measure, how to measure, and what changes will be implemented to invoke change in the system.
The auditing forms are separate. This is an event that occurred weekly during the actual implementation of the QI intervention to assess the successful role out. In the methods section under “Study of the Intervention” we have included the following paragraph to provide better clarification.
“The data collection forms were developed according to the Model for Improvement Methodology to capture the data according to the measures co-developed during the second session of our workshop assessing “How will we know that a change is an improvement?” The patient/parent report form was developed to assess the outcome measures including whether the patient experienced pain which was a simple yes/no option. This was based off a validated patient satisfaction questionnaire. [13,28] Additionally, the patient/parent was asked regarding satisfaction and was a simple Likert scale based on Likert scale development in LMIC.[29] The last 2 questions assess location of the procedure and wait time for the procedure as balancing measures. The provider self-report form included the process measures including self-assessment to ensure that each of the different interventions of the bundle were performed, and the balance measures including the success of the procedure, number of attempts, and the healthcare professional satisfaction in performing the procedure. Finally, the auditing form included the questions from both the patient/parent, and self-report form was completed one time per week on different days and time to avoid bias and to provide a third-party independent observation. In addition to the questions found on the patient/parent and self-report form, the auditing form had free text space to get open-ended feedback from the patients/parents and healthcare professionals.”
RESULTS
All along the section, medians are presented as percentages, is it correct?
This is correct. For Quality Improvement run charts the centerline is always calculated as the median. As our data being collected were percentages, the data for each run chart was the median percentage of the particular measure collected.
Page 6, Lines 214-223. This information could be used in the methods section to present the contexts in which the interventions were implemented.
Thank you for this suggestion. It has been moved to the Methods section along with a bit more information of the institutions as suggested above.
Page 7, section 3.3. Is it possible to have a table in which comfort principles, distraction techniques, and positioning techniques are described in detail?
We appreciate this suggestion. Each site used slightly different distraction techniques depending on availability of supplies such as toys, pinwheels, virtual reality headsets, etc. However, we did provide resources for common guidance regarding types of techniques and positioning techniques based on age. We have added this to the Supplemental Materials under…
Pages 10-11, Lines 320-328. How do these qualitative parts were collected and analysed? This was not described in the methods section.
Thank you for this important clarification. We have added more detail in the Methods section regarding the Auditing Form and the free text space for open-ended feedback from healthcare professionals and parents. We also have added that information in the results to describe where these quotes originated. There was no formal qualitative analysis of the free text information.
The results now state the following, “In the auditing form there were several open-ended responses on the experience of the healthcare professionals and how it impacted them.”
DISCUSSION
Page 11, From line 362. Is the sample included generalizable? The low number of infants 0-4 years could have affected the practices you have described? Moreover, a discussion about the variety of professionals performing the tasks and the potential impact on the presented results is not addressed.
Thank you for this suggestion. We agree that are ability to make generalizable statements regarding this population is difficult due to the heterogeneity of the practice settings in which it was implemented. That is why we plan to do further studies in other settings within these 4 hospitals as well as at other institutions. We have included this in our discussion. With regards to the demographics, the overall distribution of ages is fairly consistent with the age ranges of pediatric malignancy in these communities and countries, and as you mentioned it makes it difficult to formally assess the full range of interventions in the bundle, especially with regards to the infant population and ability to use breastfeeding or sucrose. We also agree regarding the fact that different professionals performed the procedures based on the context of their institution and practice. We have included the following in the limitation section “Finally, the sites implemented this initiative in different units of the hospital as well as each institution having different healthcare professionals who performed the needle-based procedure. This may have impacted generalizability. Over the coming year each site is planning for expansion and scalability to other units within the hospital to ensure sustainability in their institutions.”
I hope these suggestions will help you
These suggestions were very helpful. Thank you for your time.
Round 2
Reviewer 1 Report
Comments and Suggestions for Authors
The manuscript is improved now and authors more clearly acknowledge the limitations.
Reviewer 3 Report
Comments and Suggestions for Authors
Dear authors, thank you for allowing me to review this second version of your interesting manuscript.
I appreciated you had considered all my comments and amended the text consistently. I found this second version considerably improved, and I have no further comments to provide.
Thus, I recommend this version for publication.
Best regards